# Genetic Parameters for Age at First Calving and First Calving Interval of Beef Cattle

**DOI:** 10.3390/ani10112122

**Published:** 2020-11-16

**Authors:** Michaela Brzáková, Jindřich Čítek, Alena Svitáková, Zdeňka Veselá, Luboš Vostrý

**Affiliations:** 1Department of Genetics and Breeding of Farm Animals, Institute of Animal Science, 104 01 Prague, Czech Republic; svitakova.alena@vuzv.cz (A.S.); vesela.zdenka@vuzv.cz (Z.V.); vostry.lubos@vuzv.cz (L.V.); 2Department of Plant Production, Faculty of Agriculture, University South Bohemia, 370 05 Ceske Budejovice, Czech Republic; citek@zf.jcu.cz; 3Department of Genetics and Breeding, Faculty of Agrobiology, Food and Natural Sciences, Czech University of Life Sciences, 165 00 Prague, Czech Republic

**Keywords:** reproduction, heritability, Aberdeen Angus, Charolais, genetic correlation

## Abstract

**Simple Summary:**

Reproductive performance of beef cows affects the profitability of beef production systems. Heritability of fertility traits is generally low, which means that there is a great influence of the nongenetic environmental factors. Selection based on breeding value is an effective way to improve fertility traits, and the knowledge of genetic parameters is necessary for this approach. In this study, the two most common fertility traits—the age at first calving and first calving interval are evaluated. It was found that genetic parameters and genetic correlation between these two traits differ according to population structure (multi-breed population, the population of Aberdeen Angus, and Charolais breed). A breeding strategy should be developed within a breed.

**Abstract:**

The objective of this study was to estimate genetic parameters for age at first calving (AFC) and first calving interval (FCI) for the entire beef cattle population and separately for the Charolais (CH) and Aberdeen Angus (AA) breeds in the Czech Republic. The database of performance testing between the years 1991 and 2019 was used. The total number of cows was 83,788 from 11 breeds. After editing, the data set contained 33,533 cows, including 9321 and 4419 CH and AA cows, respectively. The relationship matrix included 85,842 animals for the entire beef population and 24,248 and 11,406 animals for the CH and AA breeds, respectively. A multibreed multitrait animal model was applied. The estimated heritability was low to moderate. Genetic correlations between AFC and FCI varied depending on the breeds from positive to negative. Differences between variance components suggest that differences between breeds should be considered before selection and breeding strategy should be developed within a breed.

## 1. Introduction

Recently, an increasing effort to include fertility traits in the genetic evaluation of beef cattle has been observed worldwide. Reproductive performance of beef cows affects the profitability of beef production systems but for a long time, the economic importance of fertility was neglected [1,2]. This could be caused by difficulties in measuring fertility compared to dairy cattle and by the low genetic heritability of fertility traits, resulting in a slow response to selection [3]. 

However, the economic value of fertility traits is considerable because poor fertility results in a low profitability in beef cattle production [4]. One of the possibilities of how to improve this production is by shortening of unproductive periods of females by reducing age at first calving and post-partum anoestrus period [5].

Age at first calving (AFC) and first calving interval (FCI) are the most mentioned traits because birth date is collected in all production systems. Moreover, AFC is a complex trait encompassing puberty and the ability to conceive, gestate, and deliver a calf [6]. Low AFC increases the productivity rate and at the same time decreases the replacement rate [7]. A calving interval of 365 days is reported as a target, but the focus should be on shortening the nonproductive periods to avoid economic losses because the length of the calving interval has a direct connection with the number of calves born annually [7]. 

This study aimed to estimate and compare genetic parameters for the early maturing breed Aberdeen Angus, later maturing breed Charolais, and the whole population of beef cattle in the Czech Republic and to assess their impact on selection for fertility traits.

## 2. Materials and Methods

Animal Care and Use Committee approval was not required for this study because the data were taken from an existing database of performance testing provided by the Czech Beef Cattle Association (Prague, Czech Republic).

### 2.1. Database

The Czech Republic is situated in a moderate climate, in the middle of Europe, where herd management with spring calving (January to April) and a pasture system with natural mating is preferred. The genetic analysis was performed for two fertility traits, age at first calving, and first calving interval. 

The database of performance testing of the Czech Beef Association was used for the analysis. Data in this database were collected routinely between the years 1991 and 2019 and are used for Czech national genetic evaluation of beef cattle for growth traits. The database consists of 463,269 animal records, including 23 breeds and their crosses. Less numerous breeds (*n* < 400 cows in breed) were excluded because of their poor connectedness within and between breeds. After this, the total number of primiparous cow records was 83,788 of 11 breeds (Table 1). 

### 2.2. Age at First Calving

Age at first calving was defined as the number of days from the cow’s birth to her first calving. For early mature breeds (Aberdeen Angus, Beef Simmental, Hereford) only AFC between 600 and 950 days was included. An AFC lower than 600 days was defined as implausible because of cows’ biological properties, and such data were removed. For Charolais, Limousine, Blonde d’Aquitaine, Gasconne and Salers only AFC higher than 880 days was used. For Galloway, Highland and Piemontese AFC between 750 and 1200 days were included. Other values were removed. The range of AFC values was chosen according to AFC distribution in population and breeding goals for these breeds in the Czech Republic. 

### 2.3. First Calving Interval 

First calving interval was defined as the number of days between the first and second calving. Other calvings were not taken into account. For genetic analysis, the first calving interval had to be between 290 and 500 days. Lower and higher values were set as missing values.

### 2.4. Data Editing

Data were edited before genetic analysis. Outlier or unreliable values, missing birth dates, and embryo transfer calves were removed from the database. Crossbreds with dairy cattle and dual-purpose breeds were also excluded. 

Then, three data sets were created. The first dataset contained all breeds (mentioned in Table 1) and their crosses and represents the entire population of beef cattle in the Czech Republic. Next, two smaller datasets were created for purebred Charolais (CH) and Aberdeen Angus (AA) only. The number of cows in these datasets was smaller compared to the numbers of AA and CH in the entire population because only pure breed individuals were included. CH and AA represent the two most numerous breeds in the Czech Republic and, at the same time, represent two extremes, later and early maturing breeds. The total number of all beef primiparous cows was 83,788 for the entire population, and for CH and AA it was 16,532 and 7444, respectively.

For AFC, all cows calved for the first time can be used, while for FCI, only cows with two calvings can be used. The range of traits was limited, others values were set as missing. For genetic parameter estimation, it was not necessary to have both AFC and FCI records and FCI could be missing. This results in different numbers of records in the analysis of these two traits. The total numbers are shown in Table 2.

### 2.5. The Connectedness

For the genetic parameters estimation, the connectedness between the animals in the dataset must be ensured. The connectedness was ensured by creating contemporary groups of animals. Two types of contemporary groups were created, herd-year-season of birth (HYSb) and herd-year-season of first calving (HYSc). Contemporary groups indirectly included the farm effect, but some farms were divided into more than one herd, which could be managed under different natural and management conditions, so the herd effects were preferred. Season was defined as a combination of three months of the year according to natural conditions: December to February, March to May, June to August, and September to November. Each HYS group included at least 5 cows fathered by at least 3 sires. For CH and AA, each HYS group included at least 5 cows fathered by at least 2 sires. These conditions were met by 33,533 cows of the whole beef population (Table 1) and by 9321 and 4419 cows of the CH and AA breeds.

### 2.6. Pedigree 

For each data set, the relationship matrix with four generations of ancestors was created. If the ancestors were unknown, a phantom parent group was assigned according to the breed of the last known animal [8]. Finally, the relationship matrix comprised 85,842 animals for the complete beef population, 24,248 animals for CH and 11,406 animals for AA.

### 2.7. Proportion of Crossbreeds 

Heterozygosity (H) was calculated as the function of the degree of heterozygosity of the cow, according to Hickey et al. [9]:*H* = *Pd*(1 − *Ps*) + *Ps*(1 − *Pd*),(1)
where *Ps* and *Pd* are the proportions of genes of the primary breed in the sire and dam, respectively. The most prevalent breed across both sire and dam pairings was considered as the primary breed for the parental pair. Table 3 shows the percentage of cows with low, medium, or high heterozygosity (0 is purebred, 1 is crossbreed). The low heterozygosity included purebred and almost purebred animals with a low impact of heterosis effect. The medium heterozygosity included animals with heterozygosity in the range between 0.25 and 0.75. The high heterozygosity included animals with heterozygosity higher than 0.75, including animals with a high impact of heterosis effect.

### 2.8. Statistical Models

The statistically significant environmental effects were determined by the procedure general linear model (GLM) in SAS 9.4 software (Cary, NC, USA) [10]. Herd-year season of birth (HYSb), herd-year-season of first calving (HYSc), herd of first calving, calving difficulty of first calving, and heterosis effect of the cow were the fixed effects. All fixed effects were tested for significance (*p* < 0.05) but also biological importance was considered. In the dataset representing the whole beef population, various breeds and their crosses were analyzed together, so the heterosis effect had been included. The effect of breed was considered in the phantom parent groups.

Age at first calving was analyzed by the following model: (2)$$Y_{ijkl}=HYSb_{i}+FHerd_{j}+b_{k}Het+Anim_{l}+e_{ijkl}$$

*Y_ijkl_* = Age at first calving in days.

$HYSb_{i}$ = Herd-year-season effect of birth (contemporary group), fixed effect, in classes.

$FHerd_{j}$ = Herd during first calving of cow, fixed effect, in classes.

$Het_{k}$ = Heterosis of cow–linear regression on heterozygosis of cow, fixed effect; the effect was included only when the whole population was tested.

Animl = Random animal additive genetic effect (4 generations included).

eijkl = Random residuum.

First calving interval was analyzed with the following model: (3)$$Y_{ijkl}=HYSc_{i}+b_{j}Het+CD_{k}+Anim_{l}+e_{ijkl}$$

*Y_ijkl_* = First calving interval in days.

$HYSc_{i}$ = Herd-year-season effect of first calving (contemporary group), fixed effect, in classes.

$Het_{j}$ = Heterosis of cow–linear regression on heterozygosis of cow, fixed effect; the effect was included only when the whole population was tested.

$CD_{k}$ = Calving difficulty of first calving, fixed effect, in classes.

Animl = Random animal additive genetic effect (4 generations included).

$e_{ijkl}$ = Random residuum.

Variance components were estimated by the Average Information Restricted Maximum Likelihood method (AIREMLF90) [11] using a general linear mixed model (a two-trait combination). The convergence criterion for the analyses was 10^−17^. 

The assumption is that animal and random residuum were distributed as N (0, A$\sigma_{u}^{2}$) and N (0, I$\sigma_{e}^{2}$), respectively, and Cov (u,e) = 0, where A is the additive relationship matrix and I is an identity matrix of order equal to the number of observations; $\sigma_{u}^{2}$ is the additive genetic variance and $\sigma_{e}^{2}$ is the error variance. 

## 3. Results and Discussion

### 3.1. Age at First Calving

The value of AFC differed according to breeds. It reflects different genetic background, herd management approaches, and the preference of individual breeders. Descriptive statistics are shown in Table 4. The average AFC for all breeds together was 996.70 ± 148 days, which corresponds to approximately 2.7 years. The lowest AFC was found for AA, the highest for CH. Comparable values of 960 days were reported by Roughsedge et al. [12]. Borman and Wilson [6] published a lower AFC of 774 days for pure bred AA, but these animals were situated only in two herds, so the effect of herd management could be considerable.

There are 11 breeds with different biological potential in the entire population. It affects the mean value of AFC and increases the standard deviation compared to the purebred population. Charolais is a late maturing breed, and therefore the majority of cows are first calved at the age of three. For AA, Czech breeders varied according to the preference of early or later calving. However, for genetic parameters estimation, only AFC between 600 and 950 was used because AA is considered as an early mature breed. 

Age at first calving is also influenced by the calving season of beef females. Calving season is often limited to several months of the year and varies worldwide according to climatic conditions and breeding management in certain countries. In Europe, including in the Czech Republic, calving season is usually situated during spring (January to April) (Table 5). Calving out of the calving season leads to problems with herd management.

The tested statistical models and included fixed effects were statistically significant at *p* < 0.0001 for all three tested groups (whole beef population, CH, and AA), except calving difficulty in AA probably because of a very low level of calving difficulties in this breed. Tested fixed effects explained approximately 81%, 42%, and 42% of all variability for the entire population, CH and AA, respectively. Most of the variability was explained by HYSb and the herd of first calving and thus the impact of herd management remained the most important effect in this case. The effect of heterosis was significant at *p* < 0.036. Cow’s heterosis reduced AFC by 7.37 days compared to purebred. The number of crossbred cows is constantly decreasing because of preferences for pure breed populations in the nucleus herds, but the proportion of crossbred cows with various levels of heterosis was still about 40%. Many crossbred cows are also included in the relationship matrix as historical ancestors.

The herd-year season of birth (HYSb) seems to be the most important effect because animals in these groups were under the same environmental, management, and nutritional conditions, which can explain a large part of the influence on fertility trait expression. Additionally, the herd of first calving was included too because in a third of cows it was different from the herd where the cow was born.

Many authors classified the contemporary group herd-year or herd-year-season as the most important effect [3,13]. The authors confirmed the high statistical significance and proportion of explained variability by this effect. AFC expression also strongly depends on the breeding season when the heifer was born and mated [14]. In contrast, a low proportion of explained variability was published by Szábo et al. [15]. The value of AFC differs in individual breeds. Dákay et al. [16] published that the effect of breed determined 97.85% of the total variance for AFC. In our analysis, the breed of cow was taken into account in groups of unknown ancestors. 

### 3.2. First Calving Interval

Descriptive statistics for FCI of the entire population and populations of CH and AA are shown in Table 4. The required length of FCI is 365 days with regard to the continuity of seasonal calving [17]. When the calving interval is longer than 365 days, the subsequent calvings during calving season may not be possible, and thus, breeders often decide to wait for the next calving season, which is connected to economic losses. Contrarily, shorter FCI can be associated with cows which calved later in the breeding season, and thus selection on short FCI may result in preference of higher age at first calving [18]. 

The population mean of FCI in this analysis was 381 days. Grossi et al. [19] published a mean of 474 days in Nellore cattle, and López-Paredes et al. [7] reported a rather long FCI of 444 days in the Blonde d´Aquitaine breed. Mean values in this analysis could be affected by the minimum and maximum restrictions, and without them, we might expect higher values. For the French population of Charolais, a higher FCI of 405 days was published [20], but the values of FCI in our study were affected by the restriction of extreme values. The AA breed is generally known for its good reproduction performance, which was proven by the lowest calving interval from our tested groups. 

The longer FCI in CH than in the whole beef population could be caused by a longer postpartum rebreeding period due to a higher frequency of calving difficulties (CD) compared to other breeds in the population. Calving difficulty at first calving is divided into four levels: an easy calving with no assistance (1), calving with slight assistance (2), difficult calving with considerable assistance (3), or Caesarean section (veterinary assistance) (4). The proportions of levels of calving difficulty are shown in Table 6. The effect of calving difficulty was tested and was significant for the whole population and for CH. For AA, this effect was not significant. This could be caused by a low level of calving difficulties in this breed. Interestingly, we found a higher frequency of calving difficulties in the cows with a higher AFC (data not shown).

Models for FCI were significant at *p* < 0.0001 for all three datasets. The tested fixed effect HYS of first calving with significance *p <* 0.0001 for all tested groups. The fixed effect of calving difficulty during first calving was with significance *p <* 0.01 for the whole dataset and *p =* 0.01 for the CH breed, respectively. The effect of calving difficulty was not significant for the AA breed (*p =* 0.41). Heterosis effect was significant at *p* < 0.03, and heterosis of dam shortened first calving interval by 3.25 days.

Finally, the model with fixed effect(s) explained 35%, 25%, and 25% of FCI variability for the whole population, Charolais and Aberdeen Angus, respectively. 

The importance of contemporary groups was stated by many authors [7,19,21]. They created a contemporary group such as combined effects herd-year-season or included other effects such as month of first calving and age at first calving, calving season or sex of calf [12,19,22]. A combination of calving difficulty and age at first calving was applied by Veselá et al. [20]. Van der Westhuizen et al. [23] confirmed that the combined effect herd-year (HY) explained the highest amount of variability and used the following effects: HY, year of calving, and age of the dam. The breeder´s decision for early calving must also be supported by the genetic potential of the heifer (fertility, direct growth ability). Before conception, heifers must be sexually mature with regular heats and reach optimal body conformation, condition, and weight [24]. However, the higher nutritional requirement (due to selection on carcass traits and larger-sized animals) can negatively affect the reproductive performance of females kept on pasture [14].

### 3.3. Genetic Parameters of Age at First Calving and First Calving Interval

Estimated genetic parameters are presented in Table 7. Generally, the heritability of AFC is low to moderate (0.18 to 0.27). This has been confirmed by our analysis and by many other authors. Martínez-Vélázquez et al. [25] and Lopez et al. [26] published heritability below 0.1. The heritability coefficient found in our study was similar to Veselá et al. [21] 0.23; Buzanskas et al. [3] 0.25; Berry and Evans [13] 0.31. Bormann and Wilson [6] estimated a heritability of 0.28 for the Aberdeen Angus and 0.17 for the Simmental breed. A higher heritability of 0.28 was reported by Martínez-Velázquez el al. [25] and Bernades et al. [27] (2015) for the Tabapua beef cattle population. Ulhôa et al. [28] published a value of 0.36. In our study, the heritability for AA was lower. Differences between heritability can be explained by differences in genetic variation between breeds or varying genotype‒environmental interaction.

The estimated heritability for the FCI was 0.08. This is comparable value with other authors: h^2^ of 0.085 [29]. Heritability of 0.02 was published by Berry and Evans [13] and Grossi et al. [19], heritability of 0.01 was reported by Lopez et al. [26], Van der Westhuizen et al. [23] and 0.05 by Ulhôa et al. [28]. A higher coefficient of 0.227 was published by Cortes et al. [30] and the value of 0.39 was found by Veselá et al. [21]. Genetic parameters for four breeds whose heritability ranged between 0.04 and 0.13 depending on the breed was estimated by Roughsedge et al. [12]. Gutierréz et al. [22] published a value of 0.12. The lowest heritability was found in the Limousine breed, h^2^ = 0.043. For the Aberdeen Angus, Simmental, and South Devon breed, the values were higher, h^2^ = 0.09, 0.125, and 0.10. For AA, we estimated a heritability 0.08. This was comparable to Roughsedge et al. [12]. These differences could be explained by differences between populations and reactions with environmental conditions.

Differences in estimated heritabilities could be caused by differences in genetic variations among breeds/populations or varying reactions to different environmental conditions. Low heritability is caused by a large unexplainable portion of residual variation [31] which could be caused by both, unknown environmental factors and unexplained additive and nonadditive genetic effects [31]. Due to the low heritability, there is an opinion that improvement in nutritional and environmental management would be more efficient than selection for calving interval traits [19]. Sure, but the effort to improve the genetic fundamentals of reproduction can never be abandoned.

### 3.4. Correlation between AFC and FCI

Phenotypic correlations between AFC and FCI were low, at 0.09, −0.26, and −0.23 for the whole population, CH and AA breeds, respectively. Some authors reported the phenotypic correlation to be not far from zero [26]. Genetic correlations varied depending on breeds. For the entire population, where early and later mature breeds are assessed together, the genetic correlation was almost zero. 

For Charolais, the genetic correlation was −0.314, i.e., the lower AFC results in a longer FCI. The negative correlation may be caused by insufficient body weight and condition in the young CH cows, which results in a longer calving interval. The level of body fat composition affects the onset of early estrus [3]. On the contrary, a higher AFC is related to adequate body weight, and the following postpartum period could be shortened because not much energy is required for their growth after calving. A negative genetic correlation for AA −0.22, Simental −0.35, and South Devon −0.71 was reported by Roughsedge et al. [12], but all of these breeds are defined as early maturing.

For the early maturing breed AA, a genetic correlation of 0.357 was found between the traits. This may be a consequence of long-term breeding for sufficient body weight before mating/conception or a preference for calving at an older age because of higher body weight. Despite this, many breeders still prefer later calvings. The reason is that in low AFC, the cow´s own growth may not be fully complete yet. More energy and nutrients are required to continue their own growth and lactation, which could prolong the postpartum period and FCI as well. 

Other studies were done on the mixed populations or on populations with only one breed. A high correlation between AFC and calving intervals was also reported [32]. A favorable genetic correlation of 0.22 was found by Berry and Evans [13] and correlation of 0.52 was reported by Lopez [26] but a negative genetic correlation (−0.25) for AFC and calving interval was found during the first 42 days of the calving season. 

## 4. Conclusions

Age at first calving and calving interval are frequently used fertility indicators in beef cattle. Low to moderate heritability of these traits was found in this analysis. Additive genetic variance varied depending on the breed. For the first calving interval, most of the variability depended on nongenetic environmental factors. For age at first calving, the genetic variance was somewhat higher. Different genetic correlations between the age at first calving and the first calving interval were found for Charolais and Aberdeen Angus, as the later and early maturing breeds, and this should be considered and taken into account in future animal breeding. Differences between variance components suggest that differences between breeds should be considered before selection and breeding strategy should be developed within a breed.

## Figures and Tables

**Table 1 animals-10-02122-t001:** Number and percentage distribution (%) of cows in the base dataset of the whole population before connectedness editing (*n*) and after connectedness editing (*n*EDIT).

Breed	*n*	%	*n* _EDIT_	%
Charolais	24,878	29.69	10,254	30.58
Beef Simmental	14,671	17.51	5313	15.84
Aberdeen Angus	13,977	16.68	5671	16.91
Hereford	10,483	12.51	5683	16.95
Limousine	8990	10.73	3318	9.89
Blonde d´Aquitaine	3250	3.88	1076	3.21
Piemontese	2740	3.27	737	2.20
Gasconne	1695	2.02	584	1.74
Galloway	1439	1.72	462	1.38
Highland	1152	1.37	284	0.85
Salers	513	0.61	151	0.45
Sum	83,788	100%	33,533	100%

**Table 2 animals-10-02122-t002:** Numbers of fertility records according to breed included in the analysis for age at first calving (AFC) and first calving interval (FCI).

Breed	AFC	FCI
Charolais	9278	7156
Beef Simmental	5061	3499
Aberdeen Angus	3932	4536
Hereford	4957	3734
Limousine	2977	2335
Blonde d´Aquitaine	943	655
Piemontese	666	500
Gasconne	438	400
Galloway	430	312
Highland	222	190
Salers	143	100
Total	29,047	23,417

**Table 3 animals-10-02122-t003:** The proportion of crossbreeds (%) in each breed.

Breed	Heterozygosity *		
	0 to 0.25	0.25 to 0.75	0.75 to 1
Charolais	74.42	9.65	15.93
Beef Simmental	60.15	13.69	26.15
Aberdeen Angus	75.68	9.89	14.42
Hereford	58.52	15.42	26.06
Limousine	67.68	11.24	21.09
Blonde d´Aquitaine	68.74	9.69	21.57
Piemontese	61.68	14.31	24.01
Gasconne	74.75	10.91	14.34
Galloway	70.67	13.62	15.71
Highland	82.20	5.21	12.59
Salers	92.40	0.39	7.21

* Heterozygosity (0 is purebred, 1 is 100% crossbred).

**Table 4 animals-10-02122-t004:** Descriptive statistics for age at first calving and first calving interval in days.

	Age at First Calving, Days					First Calving Interval, Days				
	N	Mean	SD	Min	Max	N	Mean	SD	Min	Max
**P**	29,047	996.70	148.267	600	1300	23,417	380.964	38.729	290	500
**CH**	9321	1077.77	61.064	880	1300	6204	392.175	36.725	290	500
**AA**	4419	756,133	53,653	602	950	3232	370.417	36.539	291	499

P—whole population, CH—Charolais, AA—Aberdeen Angus.

**Table 5 animals-10-02122-t005:** Frequency of first calvings during the years 1991–2019.

Month of Calving	All Beef Population		Charolais		Aberdeen Angus	
	Frequency	%	Frequency	%	Frequency	%
January	5802	17.30	2560	27.46	361	8.17
February	6661	19.86	1906	20.45	798	18.06
March	8468	25.25	1725	18.51	1343	30.39
April	4785	14.27	799	8.57	861	19.48
May	1995	5.95	253	2,71	421	9.53
June	1194	3.56	145	1.56	269	6.09
July	574	1.71	75	0.80	128	2.90
August	507	1.51	57	0.61	75	1.70
September	319	0.95	49	0.53	52	1.18
October	329	0.98	119	1.28	16	0.36
November	924	2.76	451	4.84	27	0.61
December	1975	5.89	1182	12.68	68	1.54
**Total**	33,533	100.00	9321	100.00	4419	100.00

**Table 6 animals-10-02122-t006:** Frequency of calving difficulty levels in the whole beef population and purebred Charolais and Aberdeen Angus, and the significance of the effect on the first calving interval, data after editing.

Calving Difficulty	All Beef Population		Charolais		Aberdeen Angus	
	*n*	%	*n*	%	*n*	%
1	29,175	87.00	6938	74.43	4114	93.10
2	3537	10.55	1997	21.42	220	4.98
3	755	2.25	334	3.58	75	1.70
4	66	0.20	52	0.56	10	0.23
***p* value**	*p* = 0.05		*p* < 0.01		*p* = 0.41	

1—easy calving; 2—assisted calving; 3—difficult calving; 4—Caesarean section.

**Table 7 animals-10-02122-t007:** Genetic parameters of age at first calving and first calving interval for the entire population and the populations of Charolais and Aberdeen Angus.

Model	Trait	σ^2^_a_ (SE)	cov_gen_	r_G_	σ^2^_e_ (SE)	cov_rez_	r_rez_	h^2^
Population	AFC	1337.9 (92.396)	−25.911 (30.654)	−0.074 (0.067)	3567.2 (74.266)	−297.76 (28.540)	−0.155	0.273
	FCI	90.484 (16.920)			1036.4 (17.118)			0.080
CH	AFC	569.57 (64.287)	−70.369 (24.851)	−0.314 (0.098)	1862.1 (55.380)	−357.72 (31.140)	−0.258	0.234
	FCI	85.909 (24.851)			1035.6 (28.597)			0.077
AA	AFC	347.34 (77.675)	64.297 (39.235)	0.357 (0.162)	1632.7 (69.907)	−437.48 (40.725)	−0.336	0.175
	FCI	93.225 (36.979)			1039.6 (41.590)			0.082

AFC = age at first calving, FCI = first calving interval, σ^2^_a_ = additive genetic variance; cov_a_ = additive genetic covariance; r_G_ = genetic correlation_;_ σ^2^_e_ = residual variance; cov_rez_ = residual covariance; r_rez_ = residual correlation; h^2^ = heritability (h^2^ = $\sigma_{a}^{2}/\left(\sigma_{a}^{2}+\sigma_{e}^{2}\right))$.

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
