# Peer review of "Genetic Parameters for Age at First Calving and First Calving Interval of Beef Cattle"

_animals, 2020, doi:10.3390/ani10112122_

Round 1
Reviewer 1 Report
The authors made a lot of effort to improve the paper and now it is acceptable for publication.
Author Response
Dear reviewer,
I would like to thank you for your revision. Some changes were made according to reviewer 2 in data editing, and thus the genetic parameters were reestimated. Because of this, there are changes in numbers and some changes in the text. Changes are marked in green color. Chapter 3.1 and 3.3. was edited as well. You will find it in the manuscript (green color).
Best wishes
Authors
Reviewer 2 Report
Re : Genetic parameters for age at first calving and first calving interval of beef cattle
Comments
- General comments
- I am confused after reading the response from the authours.
Was the bull allowed to run with the heifers all year around?
It depends on farm management. But most often bulls are not allowed to run with the heifers all year. This is why the effect of breeder is important.
This has serious effect on both traits. Some of the heifers cycling before the introduction of bulls will have a longer age at first calving than their genetic capabilities. If they are in a contemporary group with other late cycling heifers, they will be penalized for management error. Same applicable for FCI.
Is it not clear about the term “breeders’ decision”? How it influences your trait definition and parameter estimation?
The breeder can affect the date of bull exposure (according to maturity and weights of heifers), which influences the age at first calving of chosen cows. If some heifers have no adequate condition and weight, the breeder can wait for another breeding season (next year) and prefer for example AFC at 3 years before 2 years, for example. Each breeder has his own management and breeding strategy. These strategies are affected by the Czech Beef Cattle Association but the last decision is up to the breeder himself because he is the owner of these animals.
Breeder management is captured in the herd-year-season effect because there is an assumption that whatever strategy a breeder has, it is the same for all his cows.
Again, the management influences the traits and I am not sure how this was addressed in the evaluation.
Why different HYS and HYM groups for entire population and for the purebred Charolais and Angus population?
The connectedness in the dataset for genetic parameters estimation was ensured by creating contemporary groups (CG) of animals.
- two groups of CG were created - HYS and HYM, these groups were assigned according to explained variability in the model and also according to the importance of month in calving season.
- For entire population, HYS and HYM group included at least 5 cows fathered by at least 3 sires. For CH a AA, each HYS and HYM group included at least 5 cows fathered by at least 2 sires.
Lower requirement for number of sires for AA and CH was applied because of the lower number of animals in CH a AA population and better connectedness of one breed population without crossbreeds .
I am not entirely convinced with this answer.
What criteria was used to define the age ranges for AFC and FCI?
Ranges for AFC and FCI were defined as normal values for these traits. We need to exclude biological extremes and limit implausible data. These values were discussed with Czech Beef Breeder Association.
The figure given in the response did not support the normal distribution for AFC.
Why the same range was assumed for different breeds with different biological potentials.
Ranges for AFC and FCI were defined as normal values for these traits and only extremes and implausible data were excluded. Ranges of AFC and FCI did not affect normal distribution of traits of both breeds.
The figure given in the authors’ response, did not support the normal distribution. Furthermore, the bimodal distribution may be related breed effect. I am still not sure about the range selection.
- Results
Line 168 to 170 – How these effects influenced your trait definition and parameter estimation?
Only AFC between 600 and 1300 days and FCI between 290 and 500 days was used for genetic parameters estimation. Ranges for AFC and FCI were defined as normal values for these traits. We need to exclude biological extremes and limit implausible data. We assume that trait definition did not affect genetic parameter estimation.
In the pool estimate, animals from biologically early maturing breed and animals from biologically late maturing breed will lose more records than other breeds. Furthermore, breeders’ decision to delay the mating date also has a significant effect on the selection of range.
Need to show the effect of heterosis on AFC and FCI.
Heterosis of cow reduces age at first calving by 13,91 days and first calving interval by 3.75 days
It is not clear about the level of heterosis given in Table 2. What does the level of heterosis indicate in Table 2?
This manuscript still needs improvements. Also needs further improvement in traits selection and evaluation.
Author Response
Dear reviewer, I would like to thank you for your reminders and questions. Your comments were helpful and made me think deeply about the topics. Some changes were made in data editing, and thus the genetic parameters were reestimated. Because of this, there are changes in numbers and some changes in the text. Changes are marked in green color. Chapter 3.1 and 3.3. was edited as well. You will find it in the end of this answers or in the manuscript (green color). Here are my answers:
Question NO.1
Reviewer: Was the bull allowed to run with the heifers all year around?
Author: It depends on farm management. But most often bulls are not allowed to run with the heifers all year. This is why the effect of breeder is important.
Reviewer: This has serious effect on both traits. Some of the heifers cycling before the introduction of bulls will have a longer age at first calving than their genetic capabilities. If they are in a contemporary group with other late cycling heifers, they will be penalized for management error. Same applicable for FCI.
Author: Yes, it is true. However we are not able to statistically analyse this effect. There is no information about the type of farm management in our database, but we assumed that farmer had the same approach for all heifers and cows in his herd. Our system is very similar to the system in other countries in Europe (France, Germany, etc.). We agree that there are some management errors, but we assumed that fixed effects we used treats these errors as best as possible.
Question NO.2
Reviewer: Is it not clear about the term “breeders’ decision”? How it influences your trait definition and parameter estimation?
Author: The breeder can affect the date of bull exposure (according to maturity and weights of heifers), which influences the age at first calving of chosen cows. If some heifers have no adequate condition and weight, the breeder can wait for another breeding season (next year) and prefer for example AFC at 3 years before 2 years, for example. Each breeder has his own management and breeding strategy. These strategies are affected by the Czech Beef Cattle Association but the last decision is up to the breeder himself because he is the owner of these animals.
Breeder management is captured in the herd-year-season effect because there is an assumption that whatever strategy a breeder has, it is the same for all his cows.
Reviewer: Again, the management influences the traits and I am not sure how this was addressed in the evaluation.
Author: The range of AFC was edited according to the biological potential of breeds. There are no two peaks for breed anymore. Two peaks were made by different breeders' decisions (preference of early vs. later breeding). Now the breeder´s decision was eliminated as much as possible. The effect of breeder is also included in herd-year-season effect and fixed effect of herd during first calving of cow. Herd management is treated in HYS as fixed effect for systematic terms.
Question NO.3
Reviewer: Why different HYS and HYM groups for entire population and for the purebred Charolais and Angus population?
Author: The connectedness in the dataset for genetic parameters estimation was ensured by creating contemporary groups (CG) of animals.
- two groups of CG were created - HYS and HYM, these groups were assigned according to explained variability in the model and also according to the importance of month in calving season.
- For entire population, HYS and HYM group included at least 5 cows fathered by at least 3 sires. For CH a AA, each HYS and HYM group included at least 5 cows fathered by at least 2 sires.
Lower requirement for number of sires for AA and CH was applied because of the lower number of animals in CH a AA population and better connectedness of one breed population without crossbreeds .
Reviewer: I am not entirely convinced with this answer.
Author: The genetic parameters were reestimated using a new range of traits. At the same time, HYM of first calving was replaced by HYS of first calving. So the contemporary group of herd-years-season (the length of the season is 3 months) replaced the old contemporary group of herd-year-month of first calving.
Question NO.4
Reviewer: What criteria was used to define the age ranges for AFC and FCI?
Author: Ranges for AFC and FCI were defined as normal values for these traits. We need to exclude biological extremes and limit implausible data. These values were discussed with Czech Beef Breeder Association.
Reviewer: The figure given in the response did not support the normal distribution for AFC. Why the same range was assumed for different breeds with different biological potentials.
Ranges for AFC and FCI were defined as normal values for these traits and only extremes and implausible data were excluded. Ranges of AFC and FCI did not affect normal distribution of traits of both breeds.
The figure given in the authors’ response, did not support the normal distribution. Furthermore, the bimodal distribution may be related breed effect. I am still not sure about the range selection.
Author: Based on this comment, we edited the range of AFC for each breed.
For early mature breeds (Aberdeen Angus, Beef Simmental, Hereford) it was 600 to 950 days
For later mature large framed breeds (Charolais, Limousine, Blonde d´Aquitaine, Gasconne, and Salers) it was 880 to 1300 days. The breeding goal for Salers in the Czech Republic is to have AFC until 40 months, even if it is an early mature breed in some countries.
For Galloway, Highland and Piemontese, it was 750 to 1200 days.
Other values were removed. The range of AFC values was chosen according to AFC distribution in population, and breeding goals for these breeds in the Czech republic. Now, for each breed, the AFC values had a normal distribution (not binomal).
The range of the first calving interval (280 to 500 days) had a normal distribution, and this range was the same for all breeds. We don´t expect that there are some huge differences between breeds.
It was added in the text - chapter 2.2. Age at first calving, rows 74-82.
Question NO.5
Reviewer: Line 168 to 170 – How these effects influenced your trait definition and parameter estimation?
Author: Only AFC between 600 and 1300 days and FCI between 290 and 500 days was used for genetic parameters estimation. Ranges for AFC and FCI were defined as normal values for these traits. We need to exclude biological extremes and limit implausible data. We assume that trait definition did not affect genetic parameter estimation.
Reviewer: In the pool estimate, animals from biologically early maturing breed and animals from biologically late maturing breed will lose more records than other breeds. Furthermore, breeders’ decision to delay the mating date also has a significant effect on the selection of range.
Author: The range for AFC was reduced according breeds (see the question 4). Now the genetic parameters should be less biased.
Question NO.6
Reviewer: Need to show the effect of heterosis on AFC and FCI.
Author: Heterosis of cow reduces age at first calving by 13,91 days and first calving interval by 3.75 days
Reviewer: It is not clear about the level of heterosis given in Table 2. What does the level of heterosis indicate in Table 2?
Author: Table with the effect of heterosis was edited (Table 3). The description was changed for better understanding.
Table 3 shows the percentage of cows with low, medium, or high heterozygosity (0 is purebred, 1 is crossbreed). The low heterozygosity included purebred and almost purebred animals with a low impact of heterosis effect. The medium heterozygosity included animals with heterozygosity in the range between 0.25 to 0.75. The high heterozygosity included animals with heterozygosity higher than 0.75, including animals with a high impact of heterosis effect.
Next corections:
Chapter 3.1. Age at first calving was edited because of another approach in data editing.
Here is the old version:
The frequency of observations is affected by the seasonal calving season (spring calving) and breeder´s decisions. The distribution of AFC for both breeds shows two individual peaks, which represent consecutive years of first calving at the age of two and three years of a female. When no limit on AFC is set (AFC more than 1,300 days), a few of the primiparous cows at the age of four (third peak) are presented. These cows mostly represent cows that were not able to calve in the standard age of three years. This could be caused by reproduction or health problems, seldom by breeder decisions. Charolais is defined as a late maturing breed, and therefore the majority of cows are first calved at the age of three. For AA, Czech breeders varied according to the preference of early or later calving. Most of the cows are first calved at the age of two, the remainder at the age of three, and also a small number of animals are first calved at the age of four.
Here is the new version:
There are 11 breeds with different biological potential in the entire population. This affects the mean value of AFC and increases standard deviation compared to purebred population. Charolais is late maturing breed, and therefore the majority of cows are first calved at the age of three. For AA, Czech breeders varied according to the preference of early or later calving. However, for genetic parameters estimation, only AFC between 600 and 950 was used because AA is considered as an early mature breed.
Chapter 3.3. Genetic parameters of age at first calving and first calving interval was edited because of another approach in data editing.
Here is the old version:
Estimated genetic parameters are presented in Table 7. Generally, the heritability of AFC is low to moderate (0.18 to 0.27). This has been confirmed by our analysis and by many other authors. Martínez-Vélázquez et al. [25] and Lopez et al. [26] published heritability bellow 0.1. Heritability coefficient found in our study was similar to Veselá et al. [21] 0.23; Buzanskas et al. [3] 0.25; Berry and Evans [13] 0.31. Bormann and Wilson [6] estimated a heritability of 0.28 for the Aberdeen Angus and 0.17 for the Simmental breed. A higher heritability of 0.28 was reported by Martínez-Velázquez el al. [25] and Bernades et al. [27] (2015) for the Tabapua beef cattle population. Ulhôa et al. [28] published value of 0.36. In our study, the heritability for AA was lower. Higher heritability in CH than in AA in our analysis can be caused by many factors. First, in CH, the same AFC of 3 years is preferred by all breeders, so the breeder´s decision is not different at all, and the genetic potential is crucial. AFC of AA is more affected by breeder decisions, even if the heifer has a genetic potential to conceive earlier than other heifers in the herd. The breeder can affect the date of bull exposure according to maturity, body weight, condition of heifers, and his own preferences of early or later calvings. Therefore, the variability of AFC in this breed is caused by breeder´s decisions more than by genetic potential. Differences between heritability can also be explained by differences in genetic variation between breeds.
Here is the new version:
Estimated genetic parameters are presented in Table 7. Generally, the heritability of AFC is low to moderate (0.18 to 0.27). This has been confirmed by our analysis and by many other authors. Martínez-Vélázquez et al. [25] and Lopez et al. [26] published heritability bellow 0.1. Heritability coefficient found in our study was similar to Veselá et al. [21] 0.23; Buzanskas et al. [3] 0.25; Berry and Evans [13] 0.31. Bormann and Wilson [6] estimated a heritability of 0.28 for the Aberdeen Angus and 0.17 for the Simmental breed. A higher heritability of 0.28 was reported by Martínez-Velázquez el al. [25] and Bernades et al. [27] (2015) for the Tabapua beef cattle population. Ulhôa et al. [28] published value of 0.36. In our study, the heritability for AA was lower. Differences between heritability can be explained by differences in genetic variation between breeds or varying genotype‒environmental interaction.

This manuscript is a resubmission of an earlier submission. The following is a list of the peer review reports and author responses from that submission.
Round 1
Reviewer 1 Report
The English language should be improved by a native speaker. The present text leads sometimes to guesses for the reader and often articles are lacking.
The paper is important in the Czech Republic. But is of a lower value for an international audience because the results depend from the populations at stake.
13-14 and 39-40: Reproductive performance of beef cows affect the profitability of beef production systems.
20-21: A breeding strategy should be developed within a breed.
29-30: Please give a biological explanation later on for this difference in heritability.
68-69: For AFC all cows calved for the first time can be used. For FCI only cows with two calvings. This should result in different numbers in the analysis of these two traits.
112 and 129 and 199: How is it possible to assign cows without a pedigree to a phantom breed group? It is generally known that it is difficult to distinguish phenotypically crossbreds from purebreds.
121: Please give percentages for low, medium and high.
181: AA has a very low level of calving difficulties. That is the explanation.
185: How important is the heterosis in days?
206: first
244: affected = affect
250: our
256: How is AFC affected by the breeder in natural service systems?
287-299: Are these arguments not phenotypical explanations for genetic correlations?
Author Response
The English language should be improved by a native speaker. The present text leads sometimes to guesses for the reader and often articles are lacking.
English was edited by Elsevier Language Editing but if it is necessary to have additional english language correction we will make it. We need to ask the editor for more time for revision.
The paper is important in the Czech Republic. But is of a lower value for an international audience because the results depend from the populations at stake.
Paper is important in the Czech Republic but the approaches and conclusions may be usable internationally.
13-14 and 39-40: Reproductive performance of beef cows affect the profitability of beef production systems.
Accepted, it was changed in the text:
Row 36-37: Reproductive performance of beef cows affects the profitability of beef production systems but for a long time, the economic importance of fertility was neglected [1,2].
20-21: A breeding strategy should be developed within a breed.
Accepted.
Row 30-31.
29-30: Please give a biological explanation later on for this difference in heritability.
Accepted.
Rows: 281-298.
68-69: For AFC all cows calved for the first time can be used. For FCI only cows with two calvings. This should result in different numbers in the analysis of these two traits.
Accepted. Rows: 96-100:
... For AFC all cows calved for the first time can be used. For FCI only cows with two calvings. The range of traits was limited, others values were set as missing. For genetic parameter estimation, it was not necessary to have both, AFC and CI records and thus one of records could be missing...
112 and 129 and 199: How is it possible to assign cows without a pedigree to a phantom breed group? It is generally known that it is difficult to distinguish phenotypically crossbreds from purebreds.
There are unknown individuals in the pedigree. We don´t know anything about them (birth date, breed, etc.), but according to the breed of its progeny, we can find out or predict the breed of this unknown ancestor. Phantom parent group was assigned according to the breed of the last known animal.
Text in chapter 2.6. was edited to make it clearer.
…If the ancestors were unknown, a phantom parent group was assigned according to the breed of the last known animal [8]…
121: Please give percentages for low, medium and high.
Accepted. Percentage of crossbred females included in analysis and their level of heterosis are in the table 3.
181: AA has a very low level of calving difficulties. That is the explanation.
Accepted. It was written to the text. Rows 187-188.
185: How important is the heterosis in days?
Heterosis of cows reduces age at first calving by 13.91 days and first calving interval by 3.75 days. This finding was written to the text. Rows: 191-192 and 241-242.
206: first
Accepted
244: affected = affect
Accepted
250: our
Accepted
256: How is AFC affected by the breeder in natural service systems?
The breeder can affect the date of bull exposure (according to maturity and weights of heifers), which influences the age at first calving of chosen cows. If some heifers have no adequate condition and weight, the breeder can wait for another breeding season (next year) and prefer for example AFC at 3 years instead of 2 years, for example. Each breeder has his own management and breeding strategy. These strategies are affected by the Czech Beef Cattle Association but the final decision is up to the breeder himself because he is the owner of animals.
Breeder management is included in the herd-year-season effect because there is an assumption that whatever strategy a breeder has, it is the same for all his cows.
287-299: Are these arguments not phenotypical explanations for genetic correlations?
There are biological differences between breeds but genetic correlation between traits is important because decreasing of AFC can lead to prolonged FCI.
Reviewer 2 Report
Re : Genetic parameters for age at first calving and first calving interval of beef cattle
Comments
- General comments
- This study estimated the genetic parameters for age at first calving and first calving interval in multibreed population in the Czech Republic. Major concern in this study is the influence of seasonality and breeders’ decision on the trait definition and estimated parameters. It is not very clear how effects of seasonality and breeders’ decision were minimized in the trait construction and in the subsequent estimation of parameters.
- Abstract
- Lines 31 to 34. Not very clear and need to be rewritten
- Material and methods.
- Breeding:-
- Is it a year around mating?
- Was the bull allowed to run with the heifers all year around?
- Is it not clear about the term “breeders’ decision”? How it influences your trait definition and parameter estimation?
- Connectiveness:-
- Why different HYS and HYM groups for entire population and for the purebred Charolais and Angus population?
Data edit:-
- Why the number of animals in section 2.1 and after editing in section 2.4 were same?
Trait definition
- What criteria was used to define the age ranges for AFC and FCI?
- Why the same range was assumed for different breeds with different biological potentials.
- What proportion of animals with AFC had FCI in each breed group?
Pedigree
- Since the number of years ranged from 1991 to 2019, why the year of birth was not considered in the assignment of phantom groups?
- Results
- Line 168 to 170 – How these effects influenced your trait definition and parameter estimation?
-
Re : Genetic parameters for age at first calving and first calving interval of beef cattle
Comments
- General comments
- This study estimated the genetic parameters for age at first calving and first calving interval in multibreed population in the Czech Republic. Major concern in this study is the influence of seasonality and breeders’ decision on the trait definition and estimated parameters. It is not very clear how effects of seasonality and breeders’ decision were minimized in the trait construction and in the subsequent estimation of parameters.
- Abstract
- Lines 31 to 34. Not very clear and need to be rewritten
- Material and methods.
- Breeding:-
- Is it a year around mating?
- Was the bull allowed to run with the heifers all year around?
- Is it not clear about the term “breeders’ decision”? How it influences your trait definition and parameter estimation?
- Connectiveness:-
- Why different HYS and HYM groups for entire population and for the purebred Charolais and Angus population?
Data edit:-
- Why the number of animals in section 2.1 and after editing in section 2.4 were same?
Trait definition
- What criteria was used to define the age ranges for AFC and FCI?
- Why the same range was assumed for different breeds with different biological potentials.
- What proportion of animals with AFC had FCI in each breed group?
Pedigree
- Since the number of years ranged from 1991 to 2019, why the year of birth was not considered in the assignment of phantom groups?
- Results
- Line 168 to 170 – How these effects influenced your trait definition and parameter estimation?
- Need to show the effect of heterosis on AFC and FCI.
- Please include a table/figure to show the number of heifers calved in their 3rd and 4th years of age for different breeds.
- Line 221 It is not very clear and need to be rewritten.
- Line250 change out to our.
- Line 227 change unknon to unknown.
- Table 6. Need SE for the correlations.
- Conclusion
- Lines 313 – Not clear and need to be rewritten
This manuscript needs improvements.
Need to show the effect of heterosis on AFC and FCI. - Please include a table/figure to show the number of heifers calved in their 3rd and 4th years of age for different breeds.
- Line 221 It is not very clear and need to be rewritten.
- Line250 change out to our.
- Line 227 change unknon to unknown.
- Table 6. Need SE for the correlations.
- Conclusion
- Lines 313 – Not clear and need to be rewritten
This manuscript needs improvements.
Author Response
Dear reviewer, I would like to thank you for your reminder. Please see the attachment.
